# Study of Pressure Drops and Heat Transfer of Nonequilibrial Two-Phase Flows

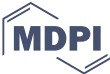

**Aleksandr V. Belyaev [1,*], Alexey V. Dedov [1], Ilya I. Krapivin [1], Aleksander N. Varava [1], Peixue Jiang [2] and Ruina Xu [2]**

[1] Department of General Physics and Nuclear Fusion, The National Research University "MPEI", Krasnokazarmennaya 14, 111250 Moscow, Russia; dedovav@mpei.ru (A.V.D.); KrapivinII@mpei.ru (I.I.K.); VaravaAN@mpei.ru (A.N.V.)

[2] Department of Thermal Engineering, Tsinghua University, Haidian District, Beijing 100084, China; jiangpx@tsinghua.edu.cn (P.J.); ruinaxu@tsinghua.edu.cn (R.X.)

\* Correspondence: BeliayevAVL@mpei.ru

**Abstract:** Currently, there are no universal methods for calculating the heat transfer and pressure drop for a wide range of two-phase flow parameters in mini-channels due to changes in the void fraction and flow regime. Many experimental studies have been carried out, and narrow-range calculation methods have been developed. With increasing pressure, it becomes possible to expand the range of parameters for applying reliable calculation methods as a result of changes in the flow regime. This paper provides an overview of methods for calculating the pressure drops and heat transfer of two-phase flows in small-diameter channels and presents a comparison of calculation methods. For conditions of high reduced pressures $p_r = p/p_{cr} \approx 0.4 \div 0.6$, the results of own experimental studies of pressure drops and flow boiling heat transfer of freons in the region of low and high mass flow rates (G = 200–2000 kg/m$^2$ s) are presented. A description of the experimental stand is given, and a comparison of own experimental data with those obtained using the most reliable calculated relations is carried out.

**Keywords:** heat transfer; hydrodynamics; high reduced pressure; flow boiling

## 1. Introduction

An important trend in the development of new energy conservation technologies is creating more miniature technical objects, an effort that requires extensive background knowledge of hydrodynamics and heat transfer in single-phase convection and flow boiling in mini-channels.

The opportunity to accurately predict the pressure drops and heat transfer and the selection of mini-channel geometry and working conditions are important factors for the design of the optimal settings of heat exchangers. In various fields of technology, one of the effective methods of heat transfer from heating surfaces is the boiling of liquid. It is necessary to experimentally confirm methods for calculating pressure drops and heat transfer.

### 1.1. Pressure Drops

The two-phase pressure drop in micro-channels is relatively high compared to conventional channels due to their very small size and relatively high mass flow rates, the latter being necessary to achieve acceptable heat transfer coefficients. Due to the high-pressure gradient, saturation temperature, and, consequently, thermophysical properties, there is a difference in pressure in mini-channels when the pressure in a certain axial position of the channel drops below the saturation pressure of the liquid, and the liquid temporarily over-

heats in this place. A pressure drop in a two-phase flow is a result of friction, acceleration, gravitation, and channel form change.

$$\Delta P_{TP} = \left[ \left( \frac{dP}{dZ} \right)_{Fr} + \left( \frac{dP}{dZ} \right)_{Ac} + \left( \frac{dP}{dZ} \right)_{Gr} \right] Z + \Delta P_F \tag{1}$$

Gravitational pressure drop is usually neglected. If the flow is adiabatic, pressure drop due to flow acceleration is neglected too. Most techniques for calculating pressure drop relate to either a homogeneous or separated flow model.

### 1.1.1. Homogeneous Equilibrium Model

For a homogeneous equilibrium model, it is assumed that liquid and gas mix with each other, and the pressure drop of a two-phase flow can be calculated using the correlations for a single-phase flow. For this, the values averaged over the entire cross-section are taken as the calculated thermophysical properties, while there is heat transfer between the phases.

$$\left( \frac{dP}{dZ} \right)_{Fr} = \left( \frac{dP}{dZ} \right)_{TP} = \xi_{TP} \frac{(G)^2}{2\rho_{TP}} \frac{1}{D_h} \tag{2}$$

where $\xi_{TP}$ is obtained by the Filonenko formula [1]:

$$\xi_{TP} = \frac{1}{\left( 1.82 \log_{10}(Re_{TP}) - 1.64 \right)^2} \tag{3}$$

$$Re_{TP} = \frac{GD_h}{\mu_{TP}} \tag{4}$$

$\mu_{TP}$ is calculated according to the method of Cicchitti et al. [2], which is the most popular method and researched for a wide range of the Reynolds numbers:

$$\mu_{TP} = x\mu_g + (1 - x)\mu_l \tag{5}$$

As it said in Zubov et al. [3], in the limit of high velocities of the mixture at high reduced pressures, there is reason to draw an analogy between the homogeneous model and the continuum model for gas flow. The traditional homogeneous flow model for shear stress on the wall uses the formula:

$$\tau_{TP} = \frac{\xi_{TP}}{8} \rho_\beta w_{TP}^2 \tag{6}$$

where $\rho_\beta = \rho'' \beta + (1 - \beta)\rho'$:

$$w_{TP} = \frac{G}{\rho'} \times \left[ 1 + x \frac{\rho' - \rho''}{\rho''} \right] \tag{7}$$

And then, pressure drops are calculated as:

$$\left( \frac{dP}{dZ} \right)_{Fr} = \frac{4\tau_{TP}}{D_h} \tag{8}$$

In two-phase flow, according to research by Venkatesan et al. [4], Cioncolini et al. [5], and Choi and Kim [6], the homogeneous flow model is applicable only to bubbly flow. Homogeneous flow conditions are fulfilled at high flow rates and mass flow rate steam content less than 0.1. At large values of the mass vapor quality $x > 0.1$, the homogeneous model, as a rule, is not applied. Under the conditions of subcooled flow, a calculation based on the homogeneous model demonstrates acceptable accuracy [7].

### 1.1.2. Separated Flow Model

Liquid and gas in the separated flow model move together, and it is taken as a fact that there is a clear phase boundary through which evaporation occurs. In the last decade,

a number of studies have been published on pressure drop in micro-channels, based on the methodology of Lockhart and Martinelli [8], who proposed using the two-phase multiplier (9) to relate the pressure drop of a two-phase flow to the pressure drop of the liquid phase:

$$\Phi_l{}^2 = \left(\frac{dP}{dZ}\right)_{TP} \bigg/ \left(\frac{dP}{dZ}\right)_1 \tag{9}$$

An example of such research is the work of Hwang et al. [9], where the working fluid was R134a, and the hydraulic diameter varied from 0.244 to 0.792 mm. The study concluded that the pressure drop increased with an increase in the Reynolds number and was similar to the pressure drop for single-phase flow in a channel with a larger equivalent diameter. In addition, the pressure drop in a two-phase flow increases with a decrease in the inner diameter. The two-phase multiplier is calculated as:

$$\Phi_l{}^2 = 1 + \frac{C}{\chi} + \frac{1}{\chi^2} \tag{10}$$

where:

$$C = 227 Re_l^{0.452} Co^{-0.82} \chi^{-0.32} \tag{11}$$

To calculate the two-phase multiplier $\Phi_l$ in conventional channels, for example, the Friedel method is often used [10], in which pipes larger than 4 mm are examined. The two-phase multiplier is calculated as:

$$\Phi_l = E + \left(\frac{3.24 FH}{Fr^{0.045} We^{0.035}}\right) \tag{12}$$

where:

$$E = (1-x)^2 + x^2 \frac{\rho_l}{\rho_g} \frac{f_g}{f_l} \tag{13}$$

$$F = x^{0.78}(1-x)^{0.224} \tag{14}$$

$$H = \left(1 - \frac{\mu_g}{\mu_l}\right)^{0.7} \left(\frac{\rho_l}{\rho_g}\right)^{0.91} \left(\frac{\mu_g}{\mu_l}\right)^{0.19} \tag{15}$$

Most formulas and methods, including the ones above, are suitable for relatively small amounts of fluid and a limited range of flow parameters and geometries. Thus, it is necessary to check the accuracy of the prediction models and select the best formula for predicting pressure drops in mini-channels.

### 1.2. Investigations of Heat Transfer in Mini-Channels

In the literature, there are many methods for determining the heat transfer coefficient when boiling a liquid flow in channels.

One of the best-known methods was by J. Chen [11], which was derived in a paper in which the boiling of a saturated water flow in a circular vertical micro-channel was investigated.

Lazarek and Black [12], when calculating the heat transfer coefficient, came to the conclusion that nucleate boiling was the main one that occurred during their tests, since the heat transfer coefficient depended on mass flow rate and heat flux density.

The experimental data in a study by Tran et al. [13] showed that when the value of vapor quality $x > 0.2$, the heat transfer coefficient does not depend on it. Here, heat transfer depended mainly on the mass velocity and not on the heat flux density. It was found that the border between the regions of the dominance of nucleate boiling and evaporation is rather abrupt and occurs at significantly smaller changes in saturation temperature than predicted.

Kenning-Cooper [14] noted that in the annular flow regime, the heat transfer coefficient is well described by J. Chen [11], but for slug flow, Chen's method gives deviations. In

addition, nucleate boiling is sensitive to surface conditions as opposed to evaporative conditions.

Gungor and Winterton [15] developed a correlation that is versatile in its application and generally gives a more accurate fit to the data than the correlations proposed by the authors of the studies reviewed. The average deviation between the calculated and measured heat transfer coefficient was 21.4% for saturated boiling and 25.0% for supercooled boiling.

Shah [16] presented, in graphical form, a general correlation called CHAPT for estimating the saturated boiling heat transfer coefficients for subcritical heating of the flow in pipes.

Liu-Winterton [17] presented a comparative analysis of the previously obtained data by J. Chen [11], Gungor and Winterton [15], and Shah [16]. In their correlation, the authors introduced the Prandtl constraint as a parameter that affects the coefficient of influence of the convective component on the heat transfer coefficient.

Kandlikar [18] conducted a comparative analysis of the earlier studies concentrating on the data obtained by the different researchers. For the correlations, data from 24 experimental studies were obtained. For comparison, the correlations of J. Chen [11], Gungor and Winterton [15], and Shah [16] were considered.

Sun and Mishima [19] conducted a comparative analysis of 13 previously obtained correlations, forming a new database. The results showed that J. Chen's correlation [11] and its modifications were not very well suited for mini-channels and that Lazarek and Black's correlation [12] was the most suitable.

Table 1 summarizes the most famous works. Obviously, the data in most of the experimental works now available in the literature were obtained for low and moderate reduced pressures [20]. In addition, the authors proposed the calculation methods, which have an empirical nature and are more suitable for describing experiments close to those described by the authors. In the field of high reduced pressures, the analysis of the literature shows a lack of researches.

**Table 1.** List of works indicating calculation methods.

| Author | Liquids | Formula |
|---|---|---|
| Tran et al. | R12, R113 | $\alpha_{TP} = 840 \left( Bo^2 We_{l\,only} \right)^{0.3} \left( \rho_g / \rho_l \right)^{0.4}$ |
| Lazarek and Black | R113 | $\alpha_{TP} = 30 Re_l^{0.857} Bo^{0.714} \frac{\lambda_l}{D_h}$ |
| Shah | R11, R12, R22 | $\alpha_{TP} = \psi \alpha_{SP}$ |
| Kenning-Cooper | Water, freons | $\alpha_{TP} = \left(1 + 1.8 X_{tt}^{-0.87}\right) \alpha_{SP}$ |
| Kandlikar | Water, R11, R12, R114, NO$_2$ | $\frac{\alpha_{TP}}{\alpha_{SP}} = \begin{cases} 1.136 Co^{-0.9} (25 Fr_l)^c + 667.2 Bo^{0.7} F_l \ (1) \\ 0.0683 Co^{-0.2} (25 Fr_l)^c + 1058 Bo^{0.7} F_l \ (2) \\ (1): Co < 0.65; \ (2): Co > 0.65 \end{cases}$ |
| Sun and Mishima | Water, freons | $\alpha_{TP} = \frac{6 Re_{lo}^{1.05} Bo^{0.54}}{We_{lo}^{0.191} \left( \frac{\rho_l}{\rho_g} \right)^{0.142}} \frac{\lambda_l}{D_h}$ |
| J. Chen | Water | $\alpha_{TP} = S\alpha_{NB} + F\alpha_{SP}$ |
| Gungor and Winterton | Water, R22, R113, R114, R11, R12 | $\alpha_{TP} = S\alpha_{NB} + F\alpha_{SP}$ |
| Liu-Winterton | Water, R113, R114, R11, R12, R22 | $\alpha_{TP} = \sqrt{(S\alpha_{NB})^2 + (F\alpha_{CB})^2}$ |

The size of the channel significantly affects the character of vaporization during flow boiling. In the region of high reduced pressures, based on the analysis performed in [21], it can be assumed that, in mini-channels, the flow regimes become identical to those seen in conventional channels. In this case, the relationships for the normal channels may be used to calculate the pressure drop and heat transfer. Based on this assumption, a method for calculating heat transfer for subcooled flow boiling in mini-channels was tested in [7].

The heat flux density was calculated as follows:

$$q = q_{boil} + q_{con} \tag{16}$$

It is assumed that convective heat transfer acts in the same way as in a single-phase turbulent flow:

$$q_{con} = \alpha_{con}(T_{wall} - T_{fluid}) \tag{17}$$

where $\alpha_{con}$ is calculated using the Petukhov formula with employees in the form [22], adjusted for the difference between the wall and liquid temperatures:

$$Nu = \frac{(\xi/8)(Re - 1000)Pr}{1 + 12.7(\xi/8)^{1/2}(Pr^{2/3} - 1)} \left(\frac{Pr_l}{Pr_{wall}}\right)^{0.25} \tag{18}$$

To calculate $q_{boil}$ in conditions of saturated flow boiling in relation (16), it is advisable to use the equation proposed by V.V. Yagov [23]:

$$q_{boil} = 3.43 \times 10^{-4} \frac{\lambda^2 \Delta T_s{}^3}{\nu \sigma T_s} \left(1 + \frac{r\Delta T}{2R_i T_s^2}\right)\left(1 + \sqrt{1 + 800B} + 400B\right) \tag{19}$$

where $B = \frac{r\left(\rho_g \frac{\mu_l}{\rho_l}\right)^{3/2}}{\sigma(\lambda T_s)^{1/2}}$ and $\Delta T_s = T_{wall} - T_s$ (all properties are determined at saturation temperature $T_s$). Modified version of Equation (19) for $q_{boil}$ for subcooled flow boiling presented in the paper [24].

## 2. Experimental Setup Description

The scheme of the experimental setup is shown in Figure 1. The hydraulic circuit allows maintaining stable flow parameters at pressures up to 2.7 MPa and temperatures up to 150 °C. A multistage centrifugal pump was used for the creation of working fluid circulation (location 6 in Figure 1). The mass flow rate was measured with a high-precision coriolis flowmeter (location 7).

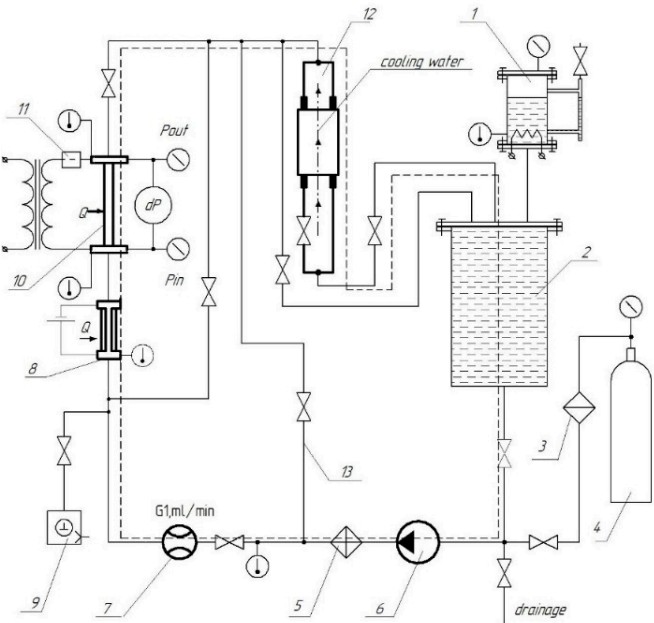

**Figure 1.** Experimental setup: (1) thermocompressor, (2) tank, (3) and (5) filters, (4) balloon with refrigerant, (6) multistage centrifugal pump, (7) coriolis flowmeter, (8) pre-heater, (9) roughing-down pump, (10) test section, (11) current transducer, (12) recuperative heat exchanger, (13) bypass line.

The working fluids in this study was R125, which has a critical temperature of 66.023 °C and a critical pressure of 3.6177 MPa. Heat capacity, heat of vaporization and critical pressure of R125 are much lower than that of water, which is very convenient for achieving the desired parameters. The working fluid was cooled by water in a recuperative heat exchanger (location 12). High reduced pressure in the circuit was created by using a

thermocompressor (location 1). A pressure sensor with a measurement accuracy of 0.2% was used for measuring pressure and pressure drops across the inlet and outlet of the test section. Chromel-Copel cable thermocouples with a cable diameter of 0.7 mm measured the inlet and outlet temperatures.

The test section was heated with alternating current. The electrical current strength was measured using an LA 55-P current transducer. The measurement error of the electric power was 1%.

The test section is shown in Figure 2. Vertical stainless-steel tubes with heated lengths of 51 mm each and internal diameters of 1 mm and 1.1 mm were used as mini-channels. The tube was electrically insulated and hydraulically sealed through PTFE (polytetrafluoroethylene) seals. Electrodes were soldered to the tube with tin.

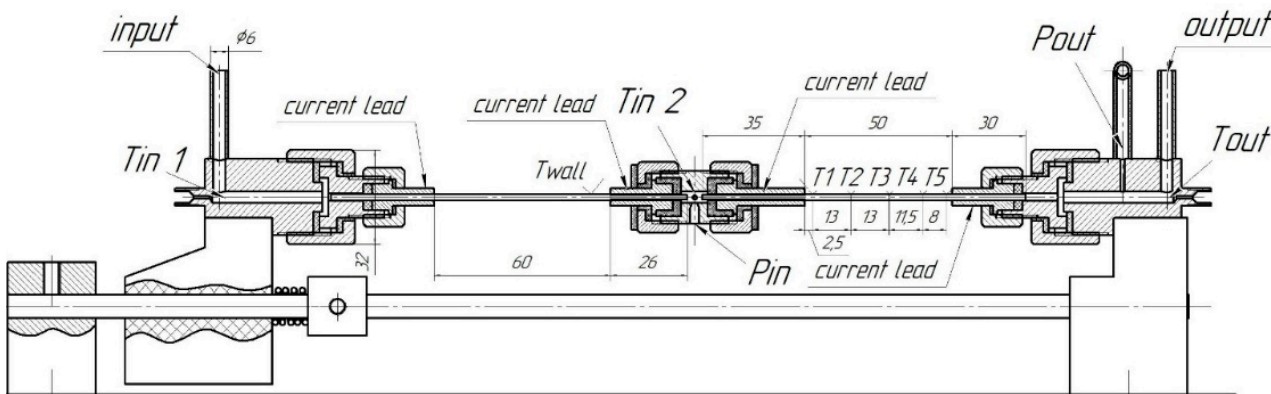

**Figure 2.** Design of the test section.

The design of the test section had temperature-compensation. The platform with the inlet collector was mounted on two vertical metal rods on which it could slide. In this way, the inlet collector had a vertical degree of freedom. The platform of the inlet collector was held by a spring along the rods towards the tube to avoid vibration and ensure the stability of the test tube.

Five Chromel-Copel thermocouples were used to take the measure values of the wall temperatures. On five cross-sections (T1–T5, see Table 2) of the working area of the tube on opposite sides of the tube diameter, the wires (diameter 0.2 mm) were welded using lasers. This mounting method for the thermocouples created low thermal inertia for the sensors and allowed the measurement of the average temperature of the wall along its perimeter. The inner wall temperatures were calculated using a correction for the wall conductivity.

**Table 2.** Coordinates of the cross-sections.

| Diameter (mm) | T1 | T2 | T3 | T4 | T5 |
|---|---|---|---|---|---|
| 1.0 | - | - | 15 | 30 | 45 |
| 1.1 | 2.5 | 15.5 | 28.5 | 40 | 48 |

## 3. Pressure Drop

In this study, experimental data on pressure drop for a range of mass flow rates $G = 200$–$2000$ kg/(m$^2$ s) were obtained at two channels with diameters 1.0 and 1.1 mm. The data were obtained for a wide range of heat flux density, which made it possible to obtain bubble and film flow regimes.

Figures 3 and 4 show the primary pressure drop data. For most of the obtained characteristics, the $\Delta p(q)$ regions of various flow regimes were observed, such as: convective heat transfer, when the pressure drop remained almost unchanged; nucleate boiling with an intense increase in pressure drop; film boiling regime, when the growth of pressure drops stopped.

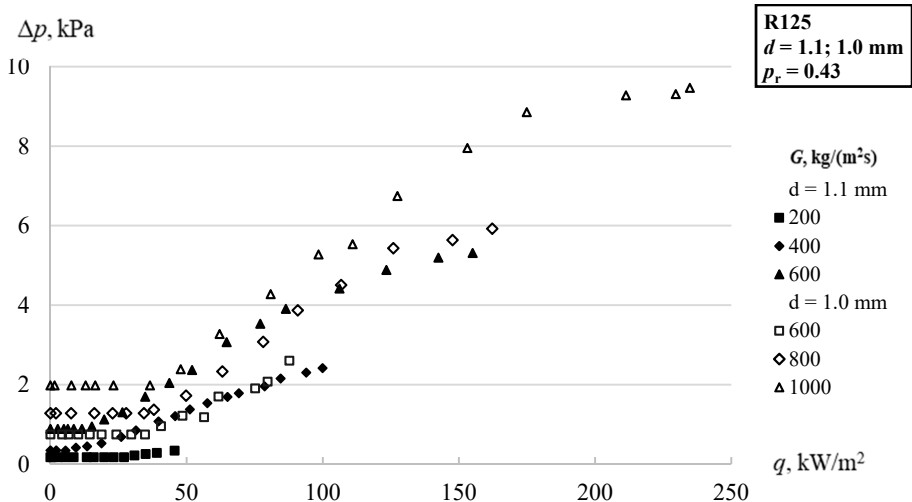

**Figure 3.** Pressure drop versus heat flux density at various values of the mass flow rates.

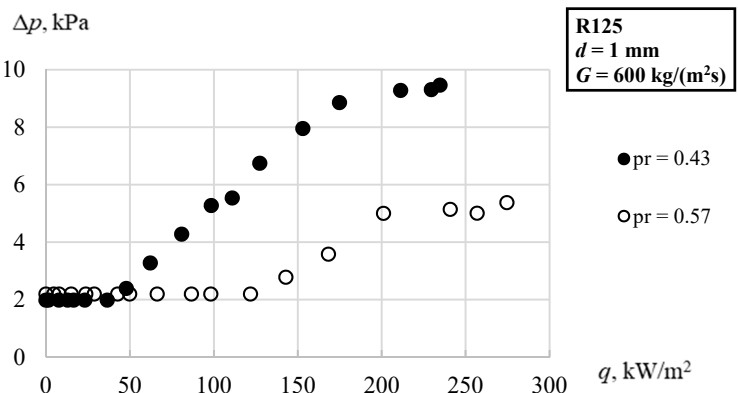

**Figure 4.** Pressure drop versus heat flux density at two values of reduced pressure.

With an increase in the diameter, the pressure drops decreased for the same values of mass flow rate (see Figure 3), which is quite natural. Analysis of the effect of reduced pressure on the pressure drops at the same values of $G = 600$ kg/(m$^2$ s) (see Figure 4) allowed us to draw the following conclusion: the flow regime changed with an increase in the reduced pressure: the region with non-increasing pressure drops due to the heat load increases from 50 to 125 kW/m$^2$.

To obtain pressure drops by calculation, the earlier described methods were used. In most of the experiments, the calculation method by [10] had too significant deviation with increasing heat flux, as can be seen in Figure 5, probably due to the quadratic dependence of the two-phase multiplier $\Phi_l$ on the vapor quality $x$. In addition, this method has been developed for channels with diameters greater than 4 mm. As a result, it was concluded that it was not suitable for generalizing the obtained data on mini-channels, even under conditions of high reduced pressures.

The analysis of the experimental data showed that the reduced pressure mostly affected the correspondence of the calculated values to the experimental data. For the investigated range of mass flow rates $G = 200$–$2000$ kg/(m$^2$ s) and values of vapor quality (up to $x \approx 0.4$) for reduced pressure $p_r = 0.43$, the best agreement with the experimental data was observed for the method of [9], which was based on a split flow model. An example of calculations for pressure $p_r = 0.43$ is shown in Figure 6.

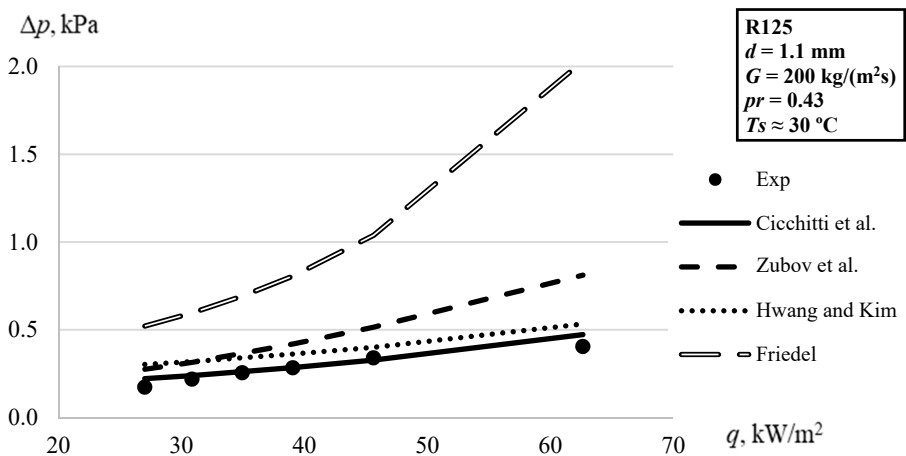

**Figure 5.** Example of the experimental data with calculated data versus heat flux.

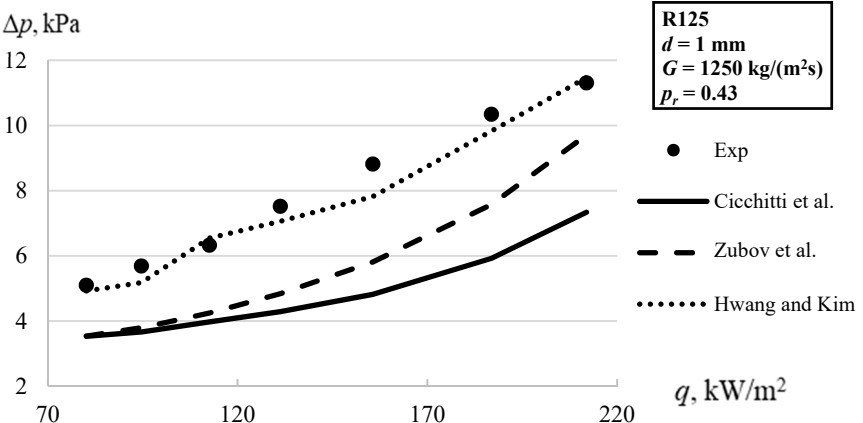

**Figure 6.** Pressure drop versus heat flux for experimental and calculated data at $G = 1250 \ \text{kg/m}^2$ s and $p_r = 0.43$.

For the data obtained at reduced pressure $p_r = 0.57$, the calculation using the homogeneous models of [2] and [3] was in better agreement with the experiment than the calculation using the split flow model. Figure 7 shows an example of a calculation for $p_r = 0.57$ and average mass flow rate $G = 750 \ \text{kg/(m}^2 \ \text{s})$.

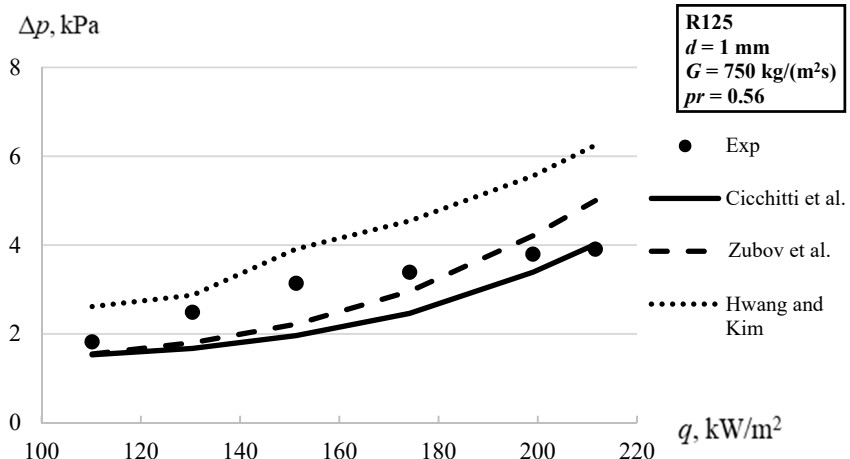

**Figure 7.** Pressure drop versus heat flux for experimental and calculated data at $G = 750 \ \text{kg/m}^2$ s and $p_r = 0.56$.

The following Table 3 presents the generalization of all experimental pressure drop data summarized by the three considered calculation methods. The data are divided into two groups according to the values of the reduced pressure.

**Table 3.** Comparison of obtained pressure drop databases with predictions of selected correlation.

| $p_r$ | Deviation | 0–10% | 0–20% | 0–30% |
|---|---|---|---|---|
| | Cicchitti et al. [2] | 14% | 23% | 37% |
| 0.43 | Zubov et al. [3] | 13% | 33% | 58% |
| | Hwang and Kim [9] | 29% | 58% | 84% |
| | Cicchitti et al. [2] | 13% | 42% | 71% |
| 0.57 | Zubov et al. [3] | 17% | 46% | 88% |
| | Hwang and Kim [9] | 0% | 4% | 13% |

From the analysis of the generalization of the obtained experimental data on pressure drop, it can be concluded that there was a significant effect of reduced pressure on the agreement of the calculated values obtained using the methods of [2,3,9] with the experimental data. As can be seen from Table 3, the homogeneous model was more suited to high reduced pressures, and the split flow model showed a good result at lower reduced pressure. This is probably due to a change in the structure of flow boiling with an increase in pressure as a result of a decrease in the diameter of the vapor bubble.

## 4. Flow Boiling Heat Transfer

Primary data on heat flux based on wall overheating relative to the saturation temperature for different mass flow rates are shown in Figure 8. At $G \leq 1750 \, \text{kg}/(\text{m}^2 \, \text{s})$, the contribution of convective heat transfer to total heat transfer was insignificant, and the boiling curves lay close to each other with a temperature deviation of about 1 °C. With increasing $G$, the contribution of convective heat transfer to total heat transfer became significant, which is quite natural, and the boiling curve for $G = 2000 \, \text{kg}/(\text{m}^2 \, \text{s})$ was significantly higher than for other points. Thus, nucleate boiling was obviously the main mechanism of heat transfer at the given mass flow rates.

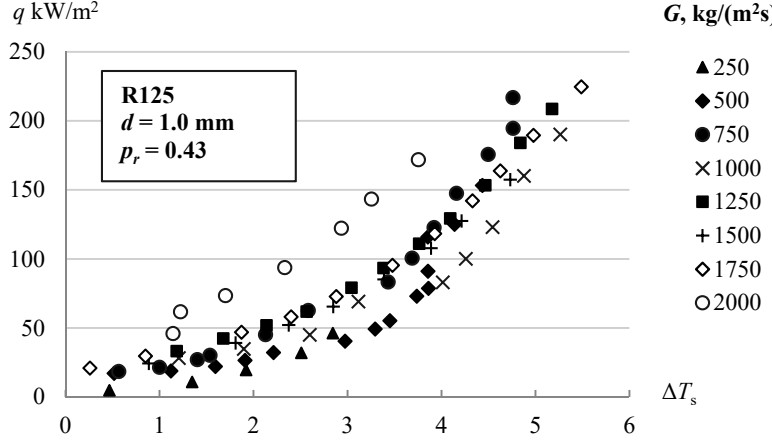

**Figure 8.** Experimental data for heat flux versus wall overheating at various mass flow rates.

The dependence of the heat transfer coefficient on the heat flux density for one mode in comparison with the calculation results given by the Petukhov Formula (18) for convective heat transfer is shown in Figure 9. It was possible to obtain a small area of convective heat transfer data points due to the impossibility of making the temperature at the entrance of the test section and, consequently, at subcooling below the room temperature. However, it can be seen from Figure 9 that the calculated values coincided with the experimental data in the region of convective heat transfer, which makes it possible to verify the experimental data.

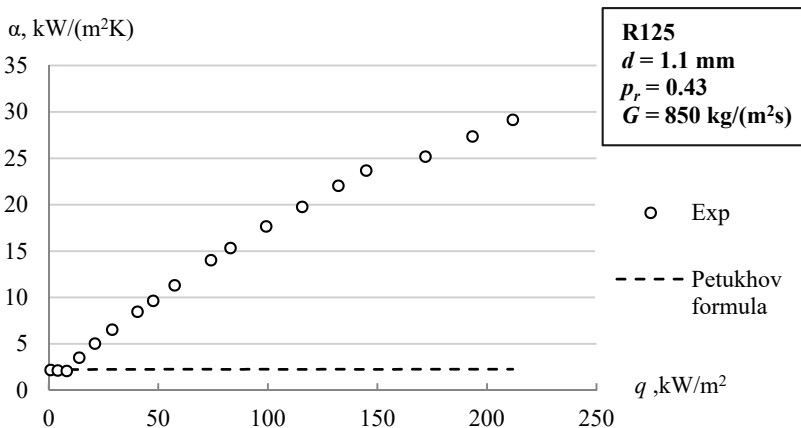

**Figure 9.** Heat transfer coefficient versus heat flux.

Comparison of the data obtained from calculations using Formulas (16)–(19) with the primary experimental data for a low mass flow rate and saturated liquid is shown in Figure 10. A comparison example of the calculation with the experimental data from [7] and [25], corresponding to moderate subcooling and high mass flow rate, is shown in Figure 11.

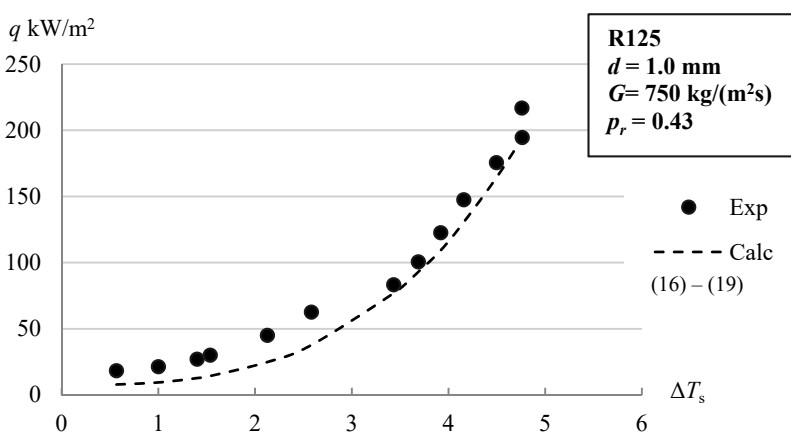

**Figure 10.** Comparison of the calculated data for heat transfer with the experimental data for saturated liquid ($x_{local} \approx 0 \div -0.4$).

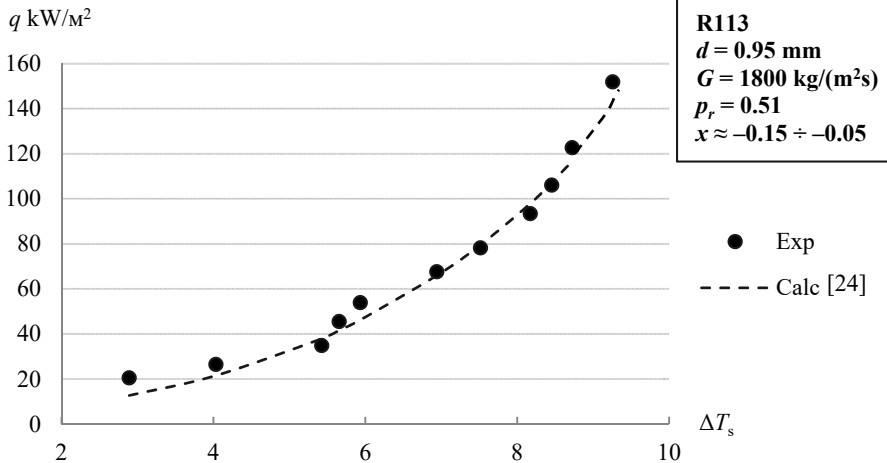

**Figure 11.** Comparison of the experimental data for subcooled liquid ([7,25], $x_{local} \approx -0.15$ to $-0.05$) with the calculated heat transfer data.

The graphs show that the calculated values were in good agreement with the experimental data. One of the features of the subcooled flow boiling, as can be seen from the primary data obtained, was a higher wall overheating (see Figure 11) compared with the saturated boiling (see Figure 10).

Figure 12 shows the data obtained in the current study for the most requested range of low and moderate mass flow rates $G = 200–1000$ kg/(m$^2$ s). Generalization was performed using Formulas (16–19). The calculation results were in good agreement with the experimental data for $x > 0$, and the mean absolute error is 16%.

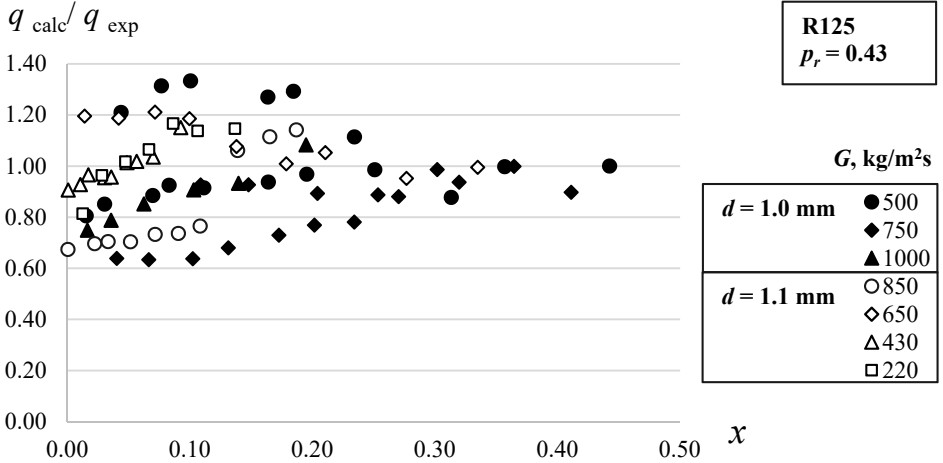

**Figure 12.** Comparison of the experimental data with the calculated data obtained using Formulas (16)–(19).

## 5. Conclusions

This paper has presented an experimental setup along with the results of an investigation of the heat transfer and pressure drop during flow boiling of R125 in two vertical channels with diameters 1.0 and 1.1 mm and lengths 51 mm each under different combinations of high reduced pressure, mass flow rate, and heat flux. These parameters were varied within the following ranges: reduced pressure $p_r \approx 0.4–0.6$, mass flow rate $G = 200–2000$ kg/(m$^2$ s), and heat flux $q$ from boiling onset to crisis.

The most popular methods in the literature for calculating pressure drop and heat transfer during flow boiling in mini-channels have been analyzed. The analysis shows a practical lack of researches with experiments at high reduced pressures.

Generalization of own data on pressure drop and heat transfer has been performed. Based on the literature review, calculation methods of [2,3,9] were chosen for the generalization of the pressure drop data. From the analysis of the generalization, it was concluded that the homogeneous model was more suited for high reduced pressures, and the split flow model showed a good result at lower reduced pressures. Thus, at a higher reduced pressure, the flow regime was more similar to the homogeneous model, whereas at a lower pressure, the flow structure was more similar to the split flow model. No such effect of the mass flow rate on flow structure was observed.

The methods for calculating pressure drop during flow boiling require further elaboration. It is necessary to establish the limits of applicability of various types of models for calculating pressure drop, depending on the reduced pressure and the degree of saturation of fluid flow.

To generalize the data on heat transfer, the previously approved method [7] was used with the division of the calculation of heat flux into convection and nucleate boiling heat flux. The presented calculation method, which are based on Formulas (16)–(19), satisfied the obtained experimental results of heat transfer with 16% of mean absolute error. This method can be applied in the most requested range of mass flow rates $G = 200–1000$ kg/m$^2$ s and $x > 0$.

**Author Contributions:** Data curation, A.V.B.; formal analysis, A.V.B.; investigation, A.V.B. and R.X.; methodology, A.V.B., A.V.D., A.N.V. and P.J.; writing—original draft preparation, A.V.B. and I.I.K.; conceptualization, A.V.D., P.J. and R.X.; project administration, A.V.D.; validation, A.V.D. and P.J.; writing—review and editing, A.V.D. and R.X.; software, I.I.K.; formal analysis, A.N.V. and P.J.; supervision, A.N.V. All authors have read and agreed to the published version of the manuscript.

**Funding:** This work was supported by RSF Grant 19-19-00410.

**Conflicts of Interest:** The authors declare no conflict of interest.

## Nomenclature

| | |
|---|---|
| $d$ | diameter, m |
| $G$ | mass flow rate, kg/(m$^2$ s) |
| $p$ | pressure, Pa |
| $T$ | temperature, K |
| $x$ | vapor quality |
| $c_p$ | specific heat, J/(kg·K) |
| $r$ | latent heat of evaporation, J/kg |
| $w$ | velocity, m/c$^2$ |

Greek symbols

| | |
|---|---|
| $\alpha$ | heat transfer coefficient, W/m$^2$·K |
| $\sigma$ | surface tension, N/m |
| $\lambda$ | thermal conductivity, W/(m·K) |
| $\xi$ | hydraulic friction factor |
| $\rho$ | density, kg/m$^3$ |
| $\beta$ | volume vapor quality |
| $\mu$ | viscosity, N·s/m$^2$ |
| $\chi$ | Martinelli parameter |
| We | Weber number $We = \frac{G^2 D_h}{\rho \sigma}$ |
| E; F; H | Friedel parameters |
| Fr | Froude number $Fr = \frac{w^2}{g D_h}$ |
| $\Phi$ | two-phase multiplier |
| Co | confinement number $Co = \left( \frac{\sigma}{g(\rho_l - \rho_g)} \right)^{0.5} D_h^{-1}$ |

Subscripts

| | |
|---|---|
| l | liquid |
| g | gas |
| boil | boiling |
| con | convective |
| calc | calculated |
| exp | experimental |
| sub | subcooled |
| cr | critical |
| in | inlet |
| s | saturated |
| r | reduced |
| Fr | friction |
| TP | two-phased |
| SP | single phase |
| CB | convective boiling |

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
