# Peer review of "Study of Pressure Drops and Heat Transfer of Nonequilibrial Two-Phase Flows"

_water, doi:10.3390/w13162275_

Round 1

Reviewer 1 Report

In the present paper, the authors compare different available calculation methods for the pressure drop and heat transfer of two-phase flows against experimental results for small-diameter channels. First, the authors recall the calculation methods, which all have an empirical nature, and describe their experimental setup. Then, they compare the experimental and calculated data and give a final conclusion.

I think that it is important to compare and validate empirical models against experiments. Here, the experimental data is in good agreement with some of the calculation methods. However, I think that the description of the methods could be improved and significance of the results could be expressed more clearly. Before the paper can be published the following comments should be taken into account in a revised version:

  • I think it would be good to really point out significance of the results more clearly. Apart from the fact that the available calculation methods were compared against new experimental data, what are the main statements and novelties of this paper?
  • Even though there is a table with the nomenclature in the back of the paper, it would be helpful for the reader to also introduce and explain the symbols within the text. Otherwise, some of the approaches described in Section 1 are difficult to comprehend without reading the original papers.
  • I like Table 1, however, I do not see a comparison of the experimental data to all the calculation methods described listed in Table 1. Why? Could you elaborate more on which of the methods are/could be applicable to your experiments and which do not?
  • Line 61: "As it said..." -> "As is said..."
  • Line 72: "Homogenous" -> "Homogeneous"
  • What are large values of the mass vapor quality? (Lines 73)
  • What is the unit of the hydraulic diameter mentioned in line 85.
  • Line 173-174: "Modified version..." This is not a complete sentence.
  • It would be good to improve the image quality of Figures 1 and 2.
  • How does the pr = 0.43 case in Fig. 4 correspond to the data plotted in Fig. 3? I would have expected to see the same as for the empty boxes in Fig. 3, but surprisingly, it seems to be the same as the empty triangles in Fig. 3.
  • I think that replacing Table 3 by a histogram would give more information to the reader.

Author Response

Thanks for the review.

Your comments were taken into account.

Pictures have been corrected.

Some of the methods from Table 1 were used for generalization in work  A.V. Belyaev, A.N. Varava, A.V. Dedov, A.T. Komov, An experimental study of flow boiling in minichannels at high reduced pressure, International Journal of Heat and Mass Transfer 110 (2017) 360–373.

Reviewer 2 Report

There is no information in the article about liquids denoted as R125, R12, R113, R11, R13, etc. The article should contain appropriate references to sources or provide comments in detail on what kind of liquids they are.

Author Response

Thanks for the review. Information about R 125 has been added